# Inference for determinantal point processes without spectral knowledge

**Rémi Bardenet**[*]
CNRS & CRIStAL
UMR 9189, Univ. Lille, France
remi.bardenet@gmail.com

**Michalis K. Titsias**[*]
Department of Informatics
Athens Univ. of Economics and Business, Greece
mtitsias@aueb.gr

[*]Both authors contributed equally to this work.

## Abstract

Determinantal point processes (DPPs) are point process models that naturally encode diversity between the points of a given realization, through a positive definite kernel $K$. DPPs possess desirable properties, such as exact sampling or analyticity of the moments, but learning the parameters of kernel $K$ through likelihood-based inference is not straightforward. First, the kernel that appears in the likelihood is not $K$, but another kernel $L$ related to $K$ through an often intractable spectral decomposition. This issue is typically bypassed in machine learning by directly parametrizing the kernel $L$, at the price of some interpretability of the model parameters. We follow this approach here. Second, the likelihood has an intractable normalizing constant, which takes the form of a large determinant in the case of a DPP over a finite set of objects, and the form of a Fredholm determinant in the case of a DPP over a continuous domain. Our main contribution is to derive bounds on the likelihood of a DPP, both for finite and continuous domains. Unlike previous work, our bounds are cheap to evaluate since they do not rely on approximating the spectrum of a large matrix or an operator. Through usual arguments, these bounds thus yield cheap variational inference and moderately expensive exact Markov chain Monte Carlo inference methods for DPPs.

## 1 Introduction

Determinantal point processes (DPPs) are point processes [1] that encode repulsiveness using algebraic arguments. They first appeared in [2], and have since then received much attention, as they arise in many fields, e.g. random matrix theory, combinatorics, quantum physics. We refer the reader to [3, 4, 5] for detailed tutorial reviews, respectively aimed at audiences of machine learners, statisticians, and probabilists. More recently, DPPs have been considered as a modelling tool, see e.g. [4, 3, 6]: DPPs appear to be a natural alternative to Poisson processes when realizations should exhibit repulsiveness. In [3], for example, DPPs are used to model diversity among summary timelines in a large news corpus. In [7], DPPs model diversity among the results of a search engine for a given query. In [4], DPPs model the spatial repartition of trees in a forest, as similar trees compete for nutrients in the ground, and thus tend to grow away from each other. With these modelling applications comes the question of learning a DPP from data, either through a parametrized form [4, 7], or non-parametrically [8, 9]. We focus in this paper on parametric inference.

Similarly to the correlation between the function values in a Gaussian process (GPs; [10]), the repulsiveness in a DPP is defined through a kernel $K$, which measures how much two points in a realization repel each other. The likelihood of a DPP involves the evaluation and the spectral decomposition of an operator $\mathcal{L}$ defined through a kernel $L$ that is related to $K$. There are two main issues that arise when performing likelihood-based inference for a DPP. First, the likelihood involves evaluating the kernel $L$, while it is more natural to parametrize $K$ instead, and there is no easy link between the parameters of these two kernels. The second issue is that the spectral decomposition of the operator $\mathcal{L}$ required in the likelihood evaluation is rarely available in practice, for computational or analytical reasons. For example, in the case of a large finite set of objects, as in the news corpus application [3], evaluating the likelihood once requires the eigendecomposition of a large matrix. Similarly, in the case of a continuous domain, as for the forest application [4], the spectral decomposition of the operator $\mathcal{L}$ may not be analytically tractable for nontrivial choices of kernel $L$. In this paper, we focus on the second issue, i.e., we provide likelihood-based inference methods that assume the kernel $L$ is parametrized, but that do not require any eigendecomposition, unlike [7]. More specifically, our main contribution is to provide bounds on the likelihood of a DPP that do not depend on the spectral decomposition of the operator $\mathcal{L}$. For the finite case, we draw inspiration from bounds used for variational inference of GPs [11], and we extend these bounds to DPPs over continuous domains.

For ease of presentation, we first consider DPPs over finite sets of objects in Section 2, and we derive bounds on the likelihood. In Section 3, we plug these bounds into known inference paradigms: variational inference and Markov chain Monte Carlo inference. In Section 4, we extend our results to the case of a DPP over a continuous domain. Readers who are only interested in the finite case, or who are unfamiliar with operator theory, can safely skip Section 4 without missing our main points. In Section 5, we experimentally validate our results, before discussing their breadth in Section 6.

## 2 DPPs over finite sets

### 2.1 Definition and likelihood

Consider a discrete set of items $\mathcal{Y} = \{\mathbf{x}_1, \ldots, \mathbf{x}_n\}$, where $\mathbf{x}_i \subset \mathbb{R}^d$ is a vector of attributes that describes item $i$. Let $K$ be a symmetric positive definite kernel [12] on $\mathbb{R}^d$, and let $\mathbf{K} = ((K(\mathbf{x}_i, \mathbf{x}_j)))$ be the Gram matrix of $K$. The DPP of kernel $K$ is defined as the probability distribution over all possible $2^n$ subsets $Y \subseteq \mathcal{Y}$ such that

$$\mathbb{P}(A \subset Y) = \det(\mathbf{K}_A), \tag{1}$$

where $\mathbf{K}_A$ denotes the sub-matrix of $\mathbf{K}$ indexed by the elements of $A$. This distribution exists and is unique if and only if the eigenvalues of $\mathbf{K}$ are in $[0,1]$ [5]. Intuitively, we can think of $K(\mathbf{x}, \mathbf{y})$ as encoding the amount of negative correlation, or "repulsiveness" between $\mathbf{x}$ and $\mathbf{y}$. Indeed, as remarked in [3], (1) first yields that diagonal elements of $\mathbf{K}$ are marginal probabilities: $\mathbb{P}(\mathbf{x}_i \in Y) = K_{ii}$. Equation (1) then entails that $\mathbf{x}_i$ and $\mathbf{x}_j$ are likely to co-occur in a realization of $Y$ if and only if

$$\det K_{\{\mathbf{x}_i, \mathbf{x}_j\}} = K(\mathbf{x}_i, \mathbf{x}_i)K(\mathbf{y}_i, \mathbf{y}_i) - K(\mathbf{x}_i, \mathbf{x}_j)^2 = \mathbb{P}(\mathbf{x}_i \in Y)\mathbb{P}(\mathbf{x}_j \in Y) - K_{ij}^2$$

is large: off-diagonal terms in $\mathbf{K}$ indicate whether points tend to co-occur.

Providing the eigenvalues of $\mathbf{K}$ are further restricted to be in $[0,1)$, the DPP of kernel $K$ has a likelihood [1]. More specifically, writing $Y_1$ for a realization of $Y$,

$$\mathbb{P}(Y = Y_1) = \frac{\det \mathbf{L}_{Y_1}}{\det(\mathbf{L} + \mathbf{I})}, \tag{2}$$

where $\mathbf{L} = (\mathbf{I} - \mathbf{K})^{-1}\mathbf{K}$, $\mathbf{I}$ is the $n \times n$ identity matrix, and $\mathbf{L}_{Y_1}$ denotes the sub-matrix of $\mathbf{L}$ indexed by the elements of $Y_1$. Now, given a realization $Y_1$, we would like to infer the parameters of kernel $K$, say the parameters $\theta_K = (a_K, \sigma_K) \in (0, \infty)^2$ of a squared exponential kernel [10]

$$K(\mathbf{x}, \mathbf{y}) = a_K \exp\left(-\frac{\|\mathbf{x} - \mathbf{y}\|^2}{2\sigma_K^2}\right). \tag{3}$$

Since the trace of $\mathbf{K}$ is the expected number of points in $Y$ [5], one can estimate $a_K$ by the number of points in the data divided by $n$ [4]. But $\sigma_K$, the repulsive "lengthscale", has to be fitted. If the number of items $n$ is large, likelihood-based methods such as maximum likelihood are too costly: each evaluation of (2) requires $\mathcal{O}(n^2)$ storage and $\mathcal{O}(n^3)$ time. Furthermore, valid choices of $\theta_K$ are constrained, since one needs to make sure the eigenvalues of $\mathbf{K}$ remain in $[0, 1)$.

A partial work-around is to note that given any symmetric positive definite kernel $L$, the likelihood (2) with matrix $\mathbf{L} = ((L(\mathbf{x}_i, \mathbf{x}_j)))$ corresponds to a valid choice of $K$, since the corresponding matrix $\mathbf{K} = \mathbf{L}(\mathbf{I}+\mathbf{L})^{-1}$ necessarily has eigenvalues in $[0, 1]$, which makes sure the DPP exists [5]. The work-around consists in directly parametrizing and inferring the kernel $L$ instead of $K$, so that the numerator of (2) is cheap to evaluate, and parameters are less constrained. Note that this step favours tractability over interpretability of the inferred parameters: if we assume $L$ to take the squared exponential form (3) instead of $K$, with parameters $a_L$ and $\sigma_L$, the number of points and the repulsiveness of the points in $Y$ do not decouple as nicely. For example, the expected number of items in $Y$ depends on $a_L$ and $\sigma_L$ now, and both parameters also significantly affect repulsiveness. There is some work investigating approximations to $\mathbf{K}$ to retain the more interpretable parametrization [4], but the machine learning literature [3, 7] almost exclusively adopts the more tractable parametrization of $L$. In this paper, we also make this choice of parametrizing $L$ directly.

Now, the computational bottleneck in the evaluation of (2) is computing $\det(\mathbf{L}+\mathbf{I})$. While this still prevents the application of maximum likelihood, bounds on this determinant can be used in a variational approach or an MCMC algorithm. In [7], bounds on $\det(\mathbf{L}+\mathbf{I})$ are proposed, requiring only the first $m$ eigenvalues of $\mathbf{L}$, where $m$ is chosen adaptively at each MCMC iteration to make the acceptance decision possible. This still requires applying power iteration methods, which are limited to finite domains, require storing the whole $n \times n$ matrix $\mathbf{L}$, and are prohibitively slow when the number of required eigenvalues $m$ is large.

## 2.2 Nonspectral bounds on the likelihood

Let us denote by $\mathbf{L}_{\mathcal{AB}}$ the submatrix of $\mathbf{L}$ where row indices correspond to the elements of $\mathcal{A}$, and column indices to those of $\mathcal{B}$. When $\mathcal{A} = \mathcal{B}$, we simply write $\mathbf{L}_{\mathcal{A}}$ for $\mathbf{L}_{\mathcal{AA}}$, and we drop the subscript when $\mathcal{A} = \mathcal{Y}$. Drawing inspiration from sparse approximations to Gaussian processes using inducing variables [11], we let $\mathcal{Z} = \{\mathbf{z}_1, \ldots, \mathbf{z}_m\}$ be an arbitrary set of points in $\mathbb{R}^d$, and we approximate $\mathbf{L}$ by $\mathbf{Q} = \mathbf{L}_{\mathcal{YZ}}[\mathbf{L}_{\mathcal{Z}}]^{-1}\mathbf{L}_{\mathcal{ZY}}$. Note that we do not constrain $\mathcal{Z}$ to belong to $\mathcal{Y}$, so that our bounds do not rely on a Nyström-type approximation [13]. We term $\mathcal{Z}$ "pseudo-inputs", or "inducing inputs".

**Proposition 1.**

$$\frac{1}{\det(\mathbf{Q}+\mathbf{I})}e^{-\text{tr}(\mathbf{L}-\mathbf{Q})} \leq \frac{1}{\det(\mathbf{L}+\mathbf{I})} \leq \frac{1}{\det(\mathbf{Q}+\mathbf{I})}. \tag{4}$$

The proof relies on a nontrivial inequality on determinants [14, Theorem 1], and is provided in the supplementary material.

# 3 Learning a DPP using bounds

In this section, we explain how to run variational inference and Markov chain Monte Carlo methods using the bounds in Proposition 1. In this section, we also make connections with variational sparse Gaussian processes more explicit.

## 3.1 Variational inference

The lower bound in Proposition 1 can be used for variational inference. Assume we have $T$ point process realizations $Y_1, \ldots, Y_T$, and we fit a DPP with kernel $L = L_\theta$. The log

likelihood can be expressed using (2)

$$\ell(\theta) = \sum_{i=1}^{T} \log \det(\mathbf{L}_{Y_t}) - T \log \det(\mathbf{L} + \mathbf{I}). \tag{5}$$

Let $\mathcal{Z}$ be an arbitrary set of $m$ points in $\mathbb{R}^d$. Proposition 1 then yields a lower bound

$$\mathcal{F}(\theta, \mathcal{Z}) \triangleq \sum_{t=1}^{T} \log \det(\mathbf{L}_{Y_t}) - T \log \det(\mathbf{Q} + \mathbf{I}) + T \mathrm{tr}\,(\mathbf{L} - \mathbf{Q}) \leq \ell(\theta). \tag{6}$$

The lower bound $\mathcal{F}(\boldsymbol{\theta}, \mathcal{Z})$ can be computed efficiently in $\mathcal{O}(nm^2)$ time, which is considerably lower than a power iteration in $\mathcal{O}(n^2)$ if $m \ll n$. Instead of maximizing $\ell(\theta)$, we thus maximize $\mathcal{F}(\theta, \mathcal{Z})$ jointly w.r.t. the kernel parameters $\theta$ and the variational parameters $\mathcal{Z}$.

To maximize (8), one can e.g. implement an EM-like scheme, alternately optimizing in $\mathcal{Z}$ and $\boldsymbol{\theta}$. Kernels are often differentiable with respect to $\theta$, and sometimes $\mathcal{F}$ will also be differentiable with respect to the pseudo-inputs $\mathcal{Z}$, so that gradient-based methods can help. In the general case, black-box optimizers such as CMA-ES [15], can also be employed.

## 3.2 Markov chain Monte Carlo inference

If approximate inference is not suitable, we can use the bounds in Proposition 1 to build a more expensive Markov chain Monte Carlo [16] sampler. Given a prior distribution $p(\theta)$ on the parameters $\theta$ of $L$, Bayesian inference relies on the posterior distribution $\pi(\theta) \propto \exp(\ell(\theta))p(\theta)$, where the log likelihood $\ell(\theta)$ is defined in (7). A standard approach to sample approximately from $\pi(\theta)$ is the Metropolis-Hastings algorithm (MH; [16, Chapter 7.3]). MH consists in building an ergodic Markov chain of invariant distribution $\pi(\theta)$. Given a proposal $q(\theta'|\theta)$, the MH algorithm starts its chain at a user-defined $\theta_0$, then at iteration $k+1$ it proposes a candidate state $\theta' \sim q(\cdot|\theta_k)$ and sets $\theta_{k+1}$ to $\theta'$ with probability

$$\alpha(\theta_k, \theta') \quad = \quad \min\left[1, \frac{e^{\ell(\theta')}p(\theta')}{e^{\ell(\theta_k)}p(\theta_k)} \frac{q(\theta_k|\theta')}{q(\theta'|\theta_k)}\right] \tag{7}$$

while $\theta_{k+1}$ is otherwise set to $\theta_k$. The core of the algorithm is thus to draw a Bernoulli variable with parameter $\alpha = \alpha(\theta, \theta')$ for $\theta, \theta' \in \mathbb{R}^d$. This is typically implemented by drawing a uniform $u \sim \mathcal{U}_{[0,1]}$ and checking whether $u < \alpha$. In our DPP application, we cannot evaluate $\alpha$. But we can use Proposition 1 to build a lower and an upper bound $\ell(\theta) \in [b_-(\theta, \mathcal{Z}), b_+(\theta, \mathcal{Z})]$, which can be arbitrarily refined by increasing the cardinality of $\mathcal{Z}$ and optimizing over $\mathcal{Z}$. We can thus build a lower and upper bound for $\alpha$

$$b_-(\theta', \mathcal{Z}') - b_+(\theta, \mathcal{Z}) + \log\left[\frac{p(\theta')}{p(\theta)}\right] \leq \log \alpha \leq b_+(\theta', \mathcal{Z}') - b_-(\theta, \mathcal{Z}) + \log\left[\frac{p(\theta')}{p(\theta)}\right]. \tag{8}$$

Now, another way to draw a Bernoulli variable with parameter $\alpha$ is to first draw $u \sim \mathcal{U}_{[0,1]}$, and then refine the bounds in (13), by augmenting the numbers $|\mathcal{Z}|, |\mathcal{Z}'|$ of inducing variables and optimizing over $\mathcal{Z}, \mathcal{Z}'$, until[1] $\log u$ is out of the interval formed by the bounds in (13). Then one can decide whether $u < \alpha$. This Bernoulli trick is sometimes named *retrospective sampling* and has been suggested as early as [17]. It has been used within MH for inference on DPPs with spectral bounds in [7], we simply adapt it to our non-spectral bounds.

## 4 The case of continuous DPPs

DPPs can be defined over very general spaces [5]. We limit ourselves here to point processes on $\mathcal{X} \subset \mathbb{R}^d$ such that one can extend the notion of likelihood. In particular, we define here a DPP on $\mathcal{X}$ as in [1, Example 5.4(c)], by defining its Janossy density. For definitions of traces and determinants of operators, we follow [18, Section VII].

### 4.1 Definition

Let $\mu$ be a measure on $(\mathbb{R}^d, \mathcal{B}(\mathbb{R}^d))$ that is continuous with respect to the Lebesgue measure, with density $\mu'$. Let $L$ be a symmetric positive definite kernel. $L$ defines a self-adjoint operator on $L^2(\mu)$ through $\mathcal{L}(f) \triangleq \int L(\mathbf{x}, \mathbf{y}) f(\mathbf{y}) d\mu(\mathbf{y})$. Assume $\mathcal{L}$ is trace-class, and

$$\mathrm{tr}(\mathcal{L}) = \int_{\mathcal{X}} L(\mathbf{x}, \mathbf{x}) d\mu(\mathbf{x}). \tag{9}$$

We assume (14) to avoid technicalities. Proving (14) can be done by requiring various assumptions on $L$ and $\mu$. Under the assumptions of Mercer's theorem, for instance, (14) will be satisfied [18, Section VII, Theorem 2.3]. More generally, the assumptions of [19, Theorem 2.12] apply to kernels over noncompact domains, in particular the Gaussian kernel with Gaussian base measure that is often used in practice. We denote by $\lambda_i$ the eigenvalues of the compact operator $\mathcal{L}$. There exists [1, Example 5.4(c)] a simple[2] point process on $\mathbb{R}^d$ such that

$$\mathbb{P}\begin{pmatrix} \text{There are } n \text{ particles, one in each of} \\ \text{the infinitesimal balls } B(\mathbf{x}_i, d\mathbf{x}_i) \end{pmatrix} = \frac{\det((L(\mathbf{x}_i, \mathbf{x}_j))}{\det(\mathcal{I} + \mathcal{L})} \mu'(x_1) \dots \mu'(x_n), \tag{10}$$

where $B(\mathbf{x}, r)$ is the open ball of center $\mathbf{x}$ and radius $r$, and where $\det(\mathcal{I} + \mathcal{L}) \triangleq \prod_{i=1}^{\infty}(\lambda_i + 1)$ is the Fredholm determinant of operator $\mathcal{L}$ [18, Section VII]. Such a process is called the determinantal point process associated to kernel $L$ and base measure $\mu$.[3] Equation (15) is the continuous equivalent of (2). Our bounds require $\boldsymbol{\Psi}$ to be computable. This is the case for the popular Gaussian kernel with Gaussian base measure.

### 4.2 Nonspectral bounds on the likelihood

In this section, we derive bounds on the likelihood (15) that do not require to compute the Fredholm determinant $\det(\mathcal{I} + \mathcal{L})$.

**Proposition 2.** *Let $\mathcal{Z} = \{\mathbf{z}_1, \dots, \mathbf{z}_m\} \subset \mathbb{R}^d$, then*

$$\frac{\det \mathbf{L}_{\mathcal{Z}}}{\det(\mathbf{L}_{\mathcal{Z}} + \boldsymbol{\Psi})} e^{-\int L(\mathbf{x}, \mathbf{x}) d\mu(\mathbf{x}) + tr(\mathbf{L}_{\mathcal{Z}}^{-1} \boldsymbol{\Psi})} \leq \frac{1}{\det(\mathcal{I} + \mathcal{L})} \leq \frac{\det \mathbf{L}_{\mathcal{Z}}}{\det(\mathbf{L}_{\mathcal{Z}} + \boldsymbol{\Psi})}, \tag{11}$$

*where $\mathbf{L}_{\mathcal{Z}} = ((L(\mathbf{z}_i, \mathbf{z}_j))$ and $\boldsymbol{\Psi}_{ij} = \int L(\mathbf{z}_i, \mathbf{x}) L(\mathbf{x}, \mathbf{z}_j) d\mu(\mathbf{x})$.*

As for Proposition 1, the proof relies on a nontrivial inequality on determinants [14, Theorem 1] and is provided in the supplementary material. We also detail in the supplementary material why (16) is the continuous equivalent to (4).

## 5 Experiments

### 5.1 A toy Gaussian continuous experiment

In this section, we consider a DPP on $\mathbb{R}$, so that the bounds derived in Section 4 apply. As in [7, Section 5.1], we take the base measure to be proportional to a Gaussian, i.e. its density is $\mu'(x) = \kappa \mathcal{N}(x|0, (2\alpha)^{-2})$. We consider a squared exponential kernel $L(x, y) = \exp\left(-\epsilon^2 \|x - y\|^2\right)$. In this particular case, the spectral decomposition of operator $\mathcal{L}$ is known [20][4]: the eigenfunctions of $\mathcal{L}$ are scaled Hermite polynomials, while the eigenvalues are a geometrically decreasing sequence. This 1D Gaussian-Gaussian example is interesting for two reasons: first, the spectral decomposition of $\mathcal{L}$ is known, so that we can sample exactly from the corresponding DPP [5] and thus generate synthetic datasets. Second, the Fredholm determinant $\det(\mathcal{I} + \mathcal{L})$ in this special case is a q-Pochhammer symbol, and

can thus be efficiently computed, see e.g. the SymPy library in Python. This allows for comparison with "ideal" likelihood-based methods, to check the validity of our MCMC sampler, for instance. We emphasize that these special properties are not needed for the inference methods in Section 3, they are simply useful to demonstrate their correctness.

We sample a synthetic dataset using $(\kappa, \alpha, \epsilon) = (1000, 0.5, 1)$, resulting in 13 points shown in red in Figure 1(a). Applying the variational inference method of Section 3.1, jointly optimizing in $\mathcal{Z}$ and $\theta = (\kappa, \alpha, \epsilon)$ using the CMA-ES optimizer [15], yields poorly consistent results: $\kappa$ varies over several orders of magnitude from one run to the other, and relative errors for $\alpha$ and $\epsilon$ go up to 100% (not shown). We thus investigate the identifiability of the parameters with the retrospective MH of Section 3.2. To limit the range of $\kappa$, we choose for $(\log \kappa, \log \alpha, \log \epsilon)$ a wide uniform prior over

$$[200, 2000] \times [-10, 10] \times [-10, 10].$$

We use a Gaussian proposal, the covariance matrix of which is adapted on-the-fly [21] so as to reach 25% of acceptance. We start each iteration with $m = 20$ pseudo-inputs, and increase it by 10 and re-optimize when the acceptance decision cannot be made. Most iterations could be made with $m = 20$, and the maximum number of inducing inputs required in our run was 80. We show the results of a run of length 10 000 in Figure 5.1. Removing a burn-in sample of size 1000, we show the resulting marginal histograms in Figures 1(b), 1(c), and 1(d). Retrospective MH and the ideal MH agree. The prior pdf is in green. The posterior marginals of $\alpha$ and $\epsilon$ are centered around the values used for simulation, and are very different from the prior, showing that the likelihood contains information about $\alpha$ and $\epsilon$. However, as expected, almost nothing is learnt about $\kappa$, as posterior and prior roughly coincide. This is an example of the issues that come with parametrizing $L$ directly, as mentioned in Section 1. It is also an example when MCMC is preferable to the variational approach in Section 3. Note that this can be detected through the variability of the results of the variational approach across independent runs.

To conclude, we show a set of optimized pseudo-inputs $\mathcal{Z}$ in black in Figure 1(a). We also superimpose the marginal of any single point in the realization, which is available through the spectral decomposition of $\mathcal{L}$ here [5]. In this particular case, this marginal is a Gaussian. Interestingly, the pseudo-inputs look like evenly spread samples from this marginal. Intuitively, they are likely to make the denominator in the likelihood (15) small, as they represent an ideal sample of the Gaussian-Gaussian DPP.

## 5.2 Diabetic neuropathy dataset

Here, we consider a real dataset of spatial patterns of nerve fibers in diabetic patients. These fibers become more clustered as diabetes progresses [22]. The dataset consists of 7 samples collected from diabetic patients at different stages of diabetic neuropathy and one healthy subject. We follow the experimental setup used in [7] and we split the total samples into two classes: Normal/Mildly Diabetic and Moderately/Severely Diabetic. The first class contains three samples and the second one the remaining four. Figure 2 displays the point process data, which contain on average 90 points per sample in the Normal/Mildly class and 67 for the Moderately/Severely class. We investigate the differences between these classes by fitting a separate DPP to each class and then quantify the differences of the repulsion or overdispersion of the point process data through the inferred kernel parameters. Paraphrasing [7], we consider a continuous DPP on $\mathbb{R}^2$, with kernel function

$$L(\mathbf{x}_i, \mathbf{x}_j) = \exp\left( -\sum_{d=1}^{2} \frac{(x_{i,d} - x_{j,d})^2}{2\sigma_d^2} \right), \tag{12}$$

and base measure proportional to a Gaussian $\mu'(\mathbf{x}) = \kappa \prod_{d=1}^{2} \mathcal{N}(x_d|\mu_d, \rho_d^2)$. As in [7], we quantify the overdispersion of realizations of such a Gaussian-Gaussian DPP through the quantities $\gamma_d = \sigma_d/\rho_d$, which are invariant to the scaling of $\mathbf{x}$. Note however that, strictly speaking, $\kappa$ also mildly influences repulsion.

We investigate the ability of the variational method in Section 3.1 to perform approximate maximum likelihood training over the kernel parameters $\theta = (\kappa, \sigma_1, \sigma_2, \rho_1, \rho_2)$. Specifically, we wish to fit a separate continuous DPP to each class by jointly maximizing the

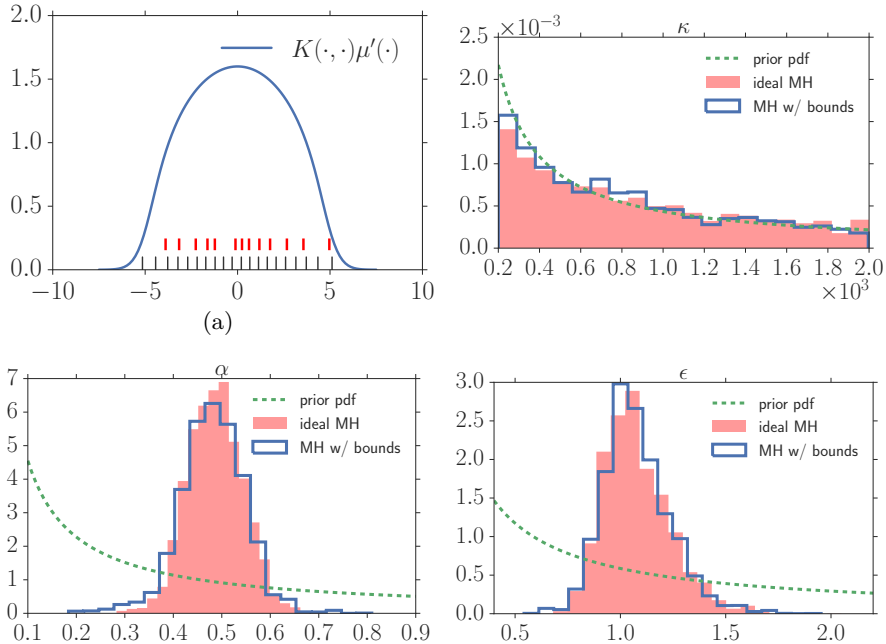

Figure 1: Results of adaptive Metropolis-Hastings in the 1D continuous experiment of Section 5.1. Figure 1(a) shows data in red, a set of optimized pseudo-inputs in black for $\theta$ set to the value used in the generation of the synthetic dataset, and the marginal of one point in the realization in blue. Figures 1(b), 1(c), and 1(d) show marginal histograms of $\kappa, \alpha, \epsilon$.

variational lower bound over $\theta$ and the inducing inputs $\mathcal{Z}$ using gradient-based optimization. Given that the number of inducing variables determines the amount of the approximation, or *compression* of the DPP model, we examine different settings for this number and see whether the corresponding trained models provide similar estimates for the overdispersion measures. Thus, we train the DPPs under different approximations having $m \in \{50, 100, 200, 400, 800, 1200\}$ inducing variables and display the estimated overdispersion measures in Figures 3(a) and 3(b). These estimated measures converge to coherent values as $m$ increases. They show a clear separation between the two classes, as also found in [7, 22]. This also suggests tuning $m$ in practice by increasing it until inference results stop varying. Furthermore, Figures 3(c) and 3(d) show the values of the upper and lower bounds on the log likelihood, which as expected, converge to the same limit as $m$ increases. We point out that the overall optimization of the variational lower bound is relatively fast in our MATLAB implementation. For instance, it takes 24 minutes for the most expensive run where $m = 1200$ to perform $1\,000$ iterations until convergence. Smaller values of $m$ yield significantly smaller times.

Finally, as in Section 5.1, we comment on the optimized pseudo-inputs. Figure 4 displays the inducing points at the end of a converged run of variational inference for various values of $m$. Similarly to Figure 1(a), these pseudo-inputs are placed in remarkably neat grids and depart significantly from their initial locations.

## 6   Discussion

We have proposed novel, cheap-to-evaluate, nonspectral bounds on the determinants arising in the likelihoods of DPPs, both finite and continuous. We have shown how to use these bounds to infer the parameters of a DPP, and demonstrated their use for expensive-but-exact MCMC and cheap-but-approximate variational inference. In particular, these bounds have some degree of freedom – the pseudo-inputs –, which we optimize so as to tighten the bounds. This optimization step is crucial for likelihood-based inference of parametric DPP models, where bounds have to adapt to the point where the likelihood is evaluated to yield

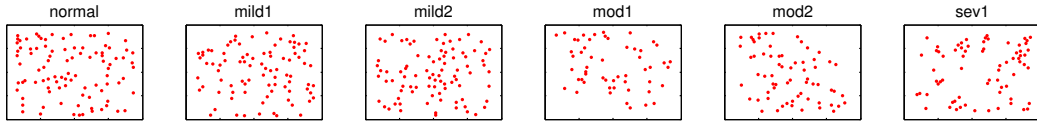

Figure 2: Six out of the seven nerve fiber samples. The first three samples (from left to right) correspond to a Normal Subject and two Mildly Diabetic Subjects, respectively. The remaining three samples correspond to a Moderately Diabetic Subject and two Severely Diabetic Subjects.

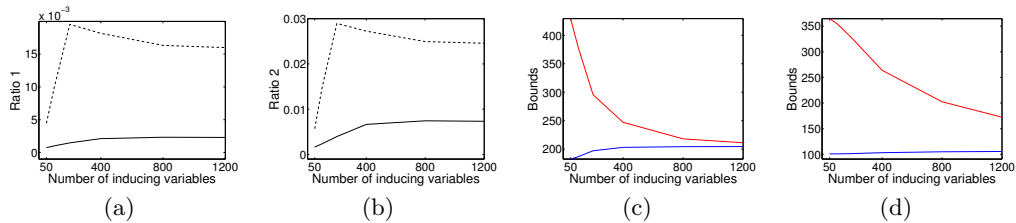

| (a) | (b) | (c) | (d) |

Figure 3: Figures 3(a) and 3(b) show the evolution of the estimated overdispersion measures $\gamma_1$ and $\gamma_2$ as functions of the number of inducing variables used. The dotted black lines correspond to the Normal/Mildly Diabetic class while the solid lines to the Moderately/Severely Diabetic class. Figure 3(c) shows the upper bound (red) and the lower bound (blue) on the log likelihood as functions of the number of inducing variables for the Normal/Mildly Diabetic class while the Moderately/Severely Diabetic case is shown in Figure 3(d).

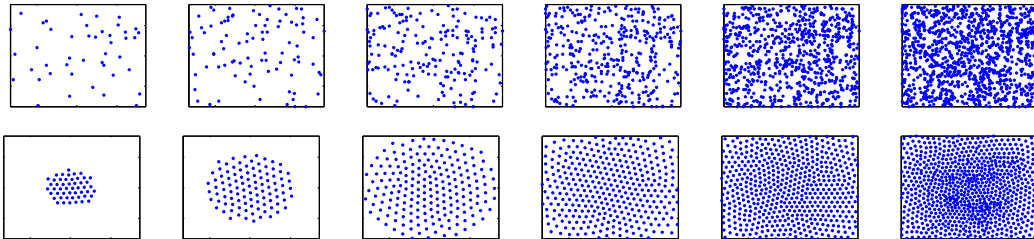

Figure 4: We illustrate the optimization over the inducing inputs $\mathcal{Z}$ for several values of $m \in \{50, 100, 200, 400, 800, 1200\}$ in the DPP of Section 5.2. We consider the Normal/Mildly diabetic class. The panels in the top row show the initial inducing input locations for various values of $m$, while the corresponding panels in the bottom row show the optimized locations.

decisions which are consistent with the ideal underlying algorithms. In future work, we plan to investigate connections of our bounds with the quadrature-based bounds for Fredholm determinants of [23]. We also plan to consider variants of DPPs that condition on the number of points in the realization, to put joint priors over the within-class distributions of the features in classification problems, in a manner related to [6]. In the long term, we will investigate connections between kernels $L$ and $K$ that could be made without spectral knowledge, to address the issue of replacing $L$ by $K$.

## Acknowledgments

We would like to thank Adrien Hardy for useful discussions and Emily Fox for kindly providing access to the diabetic neuropathy dataset. RB was funded by a research fellowship through the 2020 Science programme, funded by EPSRC grant number EP/I017909/1, and by ANR project ANR-13-BS-03-0006-01.

## Footnotes

[1]Note that this necessarily happens under fairly weak assumptions: saying that the upper and lower bounds in (4) match when $m$ goes to infinity is saying that the integral of the posterior variance of a Gaussian process with no evaluation error goes to zero as we add more distinct training points.

[2]i.e., for which all points in a realization are distinct.

[3]There is a notion of kernel $K$ for general DPPs [5], but we define $L$ directly here, for the sake of simplicity. The interpretability issues of using $L$ instead of $K$ are the same as for the finite case, see Sections 2 and 5.

[4]We follow the parametrization of [20] for ease of reference.

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
