[Supplementary Material]

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

*Proof.* The right inequality is a straightforward consequence of the Schur complement $\mathbf{L}-\mathbf{Q}$ being positive semidefinite. For instance, once could remark that for $\mathbf{v} \in \mathbb{R}^n$,

$$\mathbf{v}^T\mathbf{L}\mathbf{v} \leq 1 \Rightarrow \mathbf{v}^T\mathbf{Q}\mathbf{v} \leq 1,$$

so that

$$\int_{\mathbb{R}^n} \mathbb{1}_{\{\mathbf{v}^T(\mathbf{L}+\mathbf{I})\mathbf{v}\leq 1\}} d\mathbf{v} \leq \int_{\mathbb{R}^n} \mathbb{1}_{\{\mathbf{v}^T(\mathbf{Q}+\mathbf{I})\mathbf{v}\leq 1\}} d\mathbf{v}.$$

A change of variables using the Cholesky decompositions of $\mathbf{L} + \mathbf{I}$ and $\mathbf{Q} + \mathbf{I}$ yields the desired inequality.

The left inequality in (4) can be proved along the lines of [11], using variational arguments (as will be discussed in detail in Section 3.1.1). Next we give an alternative, more direct proof based on an inequality on determinants [14, Theorem 1]. For any real symmetric matrix $A = P\operatorname{diag}(\lambda_i)P^T$, define its absolute value as $|A| = P\operatorname{diag}(|\lambda_i|)P^T$. In particular,

for a positive semidefinite $A$, $|A| = A$. Applying [14, Theorem 1] and noting that $\mathbf{L}$, $\mathbf{Q}$ and $\mathbf{L} - \mathbf{Q}$ are positive semidefinite, it comes

$$
\begin{aligned}
\det(\mathbf{L} + \mathbf{I}) &= \det(\mathbf{L} - \mathbf{Q} + \mathbf{Q} + \mathbf{I}) \\
&\leq \det(|\mathbf{L} - \mathbf{Q}| + \mathbf{I})\det(|\mathbf{Q}| + \mathbf{I}) \\
&= \det(\mathbf{L} - \mathbf{Q} + \mathbf{I})\det(\mathbf{Q} + \mathbf{I})
\end{aligned}
\tag{5}
$$

Now, denote by $\tilde{\lambda}_i$ the eigenvalues of $\mathbf{L} - \mathbf{Q}$, which are all nonnegative. It comes

$$
\det(\mathbf{L} - \mathbf{Q} + \mathbf{I}) = \prod_{i=1}^{n}(1 + \tilde{\lambda}_i) \leq \prod_{i=1}^{n} e^{\tilde{\lambda}_i} = e^{\mathrm{tr}(\mathbf{L} - \mathbf{Q})},
\tag{6}
$$

where we used the inequality $1 + x \leq e^x$. Plugging (6) into (5) yields the left part of (4). $\quad\square$

# 3 Learning a DPP using bounds

In this section, we explain how to run variational inference and Markov chain Monte Carlo methods using the bounds in Proposition 1. In this section, we also make connections with variational sparse Gaussian processes more explicit.

## 3.1 Variational inference

The lower bound in Proposition 1 can be used for variational inference. Assume we have $T$ point process realizations $Y_1, \dots, Y_T$, and we fit a DPP with kernel $L = L_\theta$. The log likelihood can be expressed using (2)

$$
\ell(\theta) = \sum_{i=1}^{T} \log\det(\mathbf{L}_{Y_t}) - T\log\det(\mathbf{L} + \mathbf{I}).
\tag{7}
$$

Let $\mathcal{Z}$ be an arbitrary set of $m$ points in $\mathbb{R}^d$. Proposition 1 then yields a lower bound

$$
\mathcal{F}(\theta, \mathcal{Z}) \triangleq \sum_{t=1}^{T} \log\det(\mathbf{L}_{Y_t}) - T\log\det(\mathbf{Q} + \mathbf{I}) + T\mathrm{tr}\left(\mathbf{L} - \mathbf{Q}\right) \leq \ell(\theta).
\tag{8}
$$

The lower bound $\mathcal{F}(\boldsymbol{\theta}, \mathcal{Z})$ can be computed efficiently in $\mathcal{O}(nm^2)$ time, which is considerably lower than a power iteration in $\mathcal{O}(n^2)$ if $m \ll n$. Instead of maximizing $\ell(\theta)$, we thus maximize $\mathcal{F}(\theta, \mathcal{Z})$ jointly w.r.t. the kernel parameters $\theta$ and the variational parameters $\mathcal{Z}$.

To maximize (8), one can e.g. implement an EM-like scheme, alternately optimizing in $\mathcal{Z}$ and $\boldsymbol{\theta}$. Kernels are often differentiable with respect to $\theta$, and sometimes $\mathcal{F}$ will also be differentiable with respect to the pseudo-inputs $\mathcal{Z}$, so that gradient-based methods can help. In the general case, black-box optimizers such as CMA-ES [15], can also be employed.

### 3.1.1 Connections with variational inference in sparse GPs

We now discuss the connection between the variational lower bound in Proposition 1 and the variational lower bound used in sparse Gaussian process (GP; [10]) models [11]. Although we do not explore this in the current paper, this connection could extend the repertoire of variational inference algorithms for DPPs by including, for instance, stochastic optimization variants.

Assume function $f$ follows a GP distribution with zero mean function and kernel function $L$, so that the vector $\mathbf{f}$ of function values evaluated at $\mathcal{Y}$ follows the Gaussian distribution $\mathcal{N}(\mathbf{f}|\mathbf{0}, \mathbf{L})$. Then, the standard Gaussian integral yields[1]

$$
\frac{1}{\det(\mathbf{L} + \mathbf{I})} = \left(\int \mathcal{N}(\mathbf{f}|\mathbf{0}, \mathbf{L})e^{-\frac{1}{2}\mathbf{f}^T\mathbf{f}}d\mathbf{f}\right)^2.
\tag{9}
$$

Following this interpretation, we augment the vector $\mathbf{f}$ with a vector of $m$ extra auxiliary function values $\mathbf{u}$, referred to as inducing variables, evaluated at the inducing points $\mathcal{Z}$ so that jointly $(\mathbf{f}, \mathbf{u})$ follows

$$p(\mathbf{f}, \mathbf{u}) = \mathcal{N}\left(0, \begin{pmatrix} \mathbf{L} & \mathbf{L}_{\mathcal{YZ}} \\ \mathbf{L}_{\mathcal{ZY}} & \mathbf{L}_{\mathcal{Z}} \end{pmatrix}\right),$$
$$= \mathcal{N}(\mathbf{f}|\mathbf{L}_{\mathcal{YZ}}[\mathbf{L}_{\mathcal{Z}}]^{-1}\mathbf{u}, \mathbf{L} - \mathbf{Q})\mathcal{N}(\mathbf{u}|\mathbf{0}, \mathbf{L}_{\mathcal{Z}}). \tag{10}$$

Now by using the fact that $\mathcal{N}(\mathbf{f}|\mathbf{0}, \mathbf{L}) = \int p(\mathbf{f}, \mathbf{u})d\mathbf{u}$, the integral in (9) can be expanded so that

$$\frac{1}{\det(\mathbf{L} + \mathbf{I})} = \left( \int \mathcal{N}(\mathbf{f}|\mathbf{L}_{\mathcal{YZ}}[\mathbf{L}_{\mathcal{Z}}]^{-1}\mathbf{u}, \mathbf{L} - \mathbf{Q})\mathcal{N}(\mathbf{u}|\mathbf{0}, \mathbf{L}_{\mathcal{Z}})e^{-\frac{1}{2}\mathbf{f}^T\mathbf{f}}d\mathbf{f}d\mathbf{u} \right)^2. \tag{11}$$

We can bound the above integral using Jensen's inequality and the variational distribution $q(\mathbf{f}, \mathbf{u}) = \mathcal{N}(\mathbf{f}|\mathbf{L}_{\mathcal{YZ}}[\mathbf{L}_{\mathcal{Z}}]^{-1}\mathbf{u}, \mathbf{L} - \mathbf{Q})q(\mathbf{u})$, where $q(\mathbf{u})$ is a marginal variational distribution over the inducing variables $\mathbf{u}$. This form of variational distribution is exactly the one used for sparse GPs [11], and by treating the factor $q(\mathbf{u})$ optimally we can recover the left lower bound in Proposition 1, following the lines of [11]. We provide details in Appendix A.

The above connection suggests that much of the technology developed for speeding up GPs can be transferred to DPPs. For instance, if we explicitly represent the $q(\mathbf{u})$ variational distribution in the above formulation, then we can develop stochastic variational inference variants for learning DPPs based on data subsampling [?]. In other words, we can apply to DPPs stochastic variational inference algorithms for sparse GPs such as [?].

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

To see how similar (16) is to (4), we define $\mathcal{Q}_{\mathcal{Z}}$ to be the operator on $L^2(\mu)$ associated to the kernel

$$Q_{\mathcal{Z}}(\mathbf{x}, \mathbf{x}') = L(\mathbf{x}, \mathcal{Z}) \mathbf{L}_{\mathcal{Z}}^{-1} L(\mathcal{Z}, \mathbf{x}'), \tag{17}$$

where

$$L(\mathbf{x}, \mathcal{Z}) = L(\mathcal{Z}, \mathbf{x})^T \triangleq \left( L(\mathbf{x}, \mathbf{z}_1) \quad \dots \quad L(\mathbf{x}, \mathbf{z}_m) \right).$$

Then the following extension of the matrix-determinant lemma shows that the common factor in the left and right hand side of (16) is the inverse of $\det(\mathcal{I} + \mathcal{Q}_{\mathcal{Z}})$, as in (4).

**Lemma 4.1.** *With the notation of Section 4, it holds*

$$\det(\mathcal{I} + \mathcal{Q}_{\mathcal{Z}}) = \frac{\det(\mathbf{L}_{\mathcal{Z}} + \mathbf{\Psi})}{\det \mathbf{L}_{\mathcal{Z}}}.$$

*Proof.* First note that $\mathcal{Q}_{\mathcal{Z}}$ has finite rank since for $f \in L^2(\mu)$,

$$\mathcal{Q}_{\mathcal{Z}}f = \sum_{u,v=1}^{M} [\mathbf{L}_{\mathcal{Z}}^{-1}]_{uv} L(\mathbf{z}_u, \cdot) \int L(\mathbf{z}_v, y) f(y) \, d\mu(y) \in S$$

with

$$S = \text{Span}\left(L(\mathbf{z}_i, \cdot); 1 \le i \le M\right).$$

Note also that the $L(\mathbf{z}_i, \cdot)$'s are linearly independent since $L$ is a positive definite kernel. Now let $(\phi_i)_{1 \le i \le M}$ be an orthonormal basis of $S$, i.e. $\text{Span}(\phi_i; 1 \le i \le M) = S$ and

$$\int \phi_i \phi_j d\mu = \delta_{ij},$$

and define the matrix $\mathbf{W}$ by

$$\mathbf{W}_{ij} = \langle L(\cdot, \mathbf{z}_i), \phi_j \rangle,$$

where $\langle \cdot, \cdot \rangle$ denotes the inner product of $L^2(\mu)$. By definition of the Fredholm determinant for finite rank operators [18, Section VII.1 or Theorem VII.3.2], it comes

$$\det(\mathcal{I} + \mathcal{Q}_{\mathcal{Z}}) = \det\left(\left(\delta_{jk} + \langle \mathcal{Q}_{\mathcal{Z}}\phi_j, \phi_i \rangle\right)\right)_{1 \le i,j \le n}.$$

Since

$$\langle \mathcal{Q}_{\mathcal{Z}}\phi_j, \phi_i \rangle = \sum_{m,n=1}^{M} \mathbf{W}_{nj} \mathbf{W}_{mi} [\mathbf{L}_{\mathcal{Z}}^{-1}]_{mn}$$

it comes

$$\det(\mathcal{I} + \mathcal{Q}_{Z}) = \det(\mathbf{I} + \mathbf{W}^T \mathbf{L}_{\mathcal{Z}}^{-1} \mathbf{W}).$$

Applying the classical matrix determinant lemma, it comes

$$\det(\mathcal{I} + \mathcal{Q}_{\mathcal{Z}}) = \frac{\det(\mathbf{L}_{\mathcal{Z}} + \mathbf{W}\mathbf{W}^T)}{\det \mathbf{L}_{\mathcal{Z}}}.$$

We finally remark that

$$\begin{aligned}
[\mathbf{W}\mathbf{W}^T]_{ij} &= \sum_{k=1}^{M} \langle L(\mathbf{z}_i, \cdot), \phi_k \rangle \langle L(\mathbf{z}_j, \cdot), \phi_k \rangle \\
&= \left\langle \sum_{k=1}^{M} \langle L(\mathbf{z}_i, \cdot), \phi_k \rangle \phi_k, \, L(\mathbf{z}_j, \cdot) \right\rangle \\
&= \langle L(\mathbf{z}_i, \cdot), \, L(\mathbf{z}_j, \cdot) \rangle.
\end{aligned}$$

$\square$

*Proof.* (of Proposition 2) We first prove the right inequality in (16). From [18, Section VII.7], using (14), it holds

$$\det(\mathcal{I} + \mathcal{L}) = 1 + \sum_{k=1}^{\infty} \frac{1}{k!} \int \det\left(\left(L(\mathbf{x}_i, \mathbf{x}_j)\right)\right) d\mu(\mathbf{x}_1) \dots d\mu(\mathbf{x}_k). \tag{18}$$

We now apply the same argument as in the proof of the finite case (proof of Proposition 1). Denoting $\mathbf{L}_{\mathcal{X}\mathcal{Y}} = \det((L(\mathbf{x}_i, \mathbf{y}_i)))$ and $\mathbf{L}_{\mathcal{X}} = ((\det L(\mathbf{x}_i, \mathbf{x}_j))$, we know from the positive definiteness of the kernel $L$ that $\mathbf{L}_{\mathcal{X}} - \mathbf{L}_{\mathcal{X}\mathcal{Z}}\mathbf{L}_{\mathcal{Z}}^{-1}\mathbf{L}_{\mathcal{Z}\mathcal{X}}$ is positive semidefinite, which yields

$$\det \mathbf{L}_{\mathcal{X}} \ge \det \mathbf{L}_{\mathcal{X}\mathcal{Z}}\mathbf{L}_{\mathcal{Z}}^{-1}\mathbf{L}_{\mathcal{Z}\mathcal{X}}.$$

Plugging this into (18) yields the right inequality in (16).

Upon noting that

$$\text{tr}(\mathcal{Q}_{\mathcal{Z}}) = \sum_{mn} [\mathbf{L}_{\mathcal{Z}}^{-1}]_{uv} \int L(\mathbf{x}, \mathbf{z}_u) L(\mathbf{z}_v, \mathbf{x}) d\mu(\mathbf{x}) = \text{tr}(\mathbf{L}_{\mathcal{Z}}^{-1}\mathbf{\Psi}),$$

the proof of the left inequality in (16) follows the lines of the proof of Proposition 1, since the main tool [14, Theorem 1] is valid for any trace-class operators. $\square$

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

## A    On the connection to variational sparse GPs

Here, we provide further details about how the variational lower bound for DPPs over finite sets in Proposition 1 can be obtained by the variational approach to sparse GPs [11]. As mentioned in Section 3.1.1, it holds that

$$\frac{1}{\det(\mathbf{L} + \mathbf{I})} = \left( \int \mathcal{N}(\mathbf{f}|\mathbf{L}_{\mathcal{Y}\mathcal{Z}}[\mathbf{L}_{\mathcal{Z}}]^{-1}\mathbf{u}, \mathbf{L} - \mathbf{Q})\mathcal{N}(\mathbf{u}|\mathbf{0}, \mathbf{L}_{\mathcal{Z}})e^{-\frac{1}{2}\mathbf{f}^T\mathbf{f}}d\mathbf{f}d\mathbf{u} \right)^2 .$$

Taking logarithms yields

$$\log \frac{1}{\det(\mathbf{L} + \mathbf{I})} = 2\log \int \mathcal{N}(\mathbf{f}|\mathbf{L}_{\mathcal{Y}\mathcal{Z}}[\mathbf{L}_{\mathcal{Z}}]^{-1}\mathbf{u}, \mathbf{L} - \mathbf{Q})\mathcal{N}(\mathbf{u}|\mathbf{0}, \mathbf{L}_{\mathcal{Z}})e^{-\frac{1}{2}\mathbf{f}^T\mathbf{f}}d\mathbf{f}d\mathbf{u}.$$

A lower bound to the likelihood (2) can thus be obtained if we bound

$$\mathcal{F} = \log \int \mathcal{N}(\mathbf{f}|\mathbf{L}_{\mathcal{Y}\mathcal{Z}}[\mathbf{L}_{\mathcal{Z}}]^{-1}\mathbf{u}, \mathbf{L} - \mathbf{Q})\mathcal{N}(\mathbf{u}|\mathbf{0}, \mathbf{L}_{\mathcal{Z}})e^{-\frac{1}{2}\mathbf{f}^T\mathbf{f}}d\mathbf{f}d\mathbf{u}.$$

This has a similar functional form with the marginal likelihood in a standard GP regression model: $e^{-\frac{1}{2}\mathbf{f}^T\mathbf{f}}$ plays the role of an unnormalized Gaussian likelihood where the observation vector is equal to zero and the noise variance is equal to one. To lower bound the above we can consider the variational distribution $q(\mathbf{f}, \mathbf{u}) = \mathcal{N}(\mathbf{f}|\mathbf{L}_{\mathcal{Y}\mathcal{Z}}[\mathbf{L}_{\mathcal{Z}}]^{-1}\mathbf{u}, \mathbf{L} - \mathbf{Q})q(\mathbf{u})$ and apply Jensen's inequality so that

$$\mathcal{F} \geq \int \mathcal{N}(\mathbf{f}|\mathbf{L}_{\mathcal{Y}\mathcal{Z}}[\mathbf{L}_{\mathcal{Z}}]^{-1}\mathbf{u}, \mathbf{L} - \mathbf{Q})q(\mathbf{u}) \log \frac{\mathcal{N}(\mathbf{u}|\mathbf{0}, \mathbf{L}_{\mathcal{Z}})e^{-\frac{1}{2}\mathbf{f}^T\mathbf{f}}}{q(\mathbf{u})}d\mathbf{f}d\mathbf{u},$$

where the term $\mathcal{N}(\mathbf{f}|\mathbf{L}_{\mathcal{Y}\mathcal{Z}}[\mathbf{L}_{\mathcal{Z}}]^{-1}\mathbf{u}, \mathbf{L} - \mathbf{Q})$ cancels out inside the logarithm. This can be written as

$$\mathcal{F} \geq \int q(\mathbf{u}) \left\{ -\frac{1}{2} \int \mathcal{N}(\mathbf{f}|\mathbf{L}_{\mathcal{Y}\mathcal{Z}}[\mathbf{L}_{\mathcal{Z}}]^{-1}\mathbf{u}, \mathbf{L} - \mathbf{Q})\mathbf{f}^T\mathbf{f}d\mathbf{f} + \log \frac{\mathcal{N}(\mathbf{u}|\mathbf{0}, \mathbf{L}_{\mathcal{Z}})}{q(\mathbf{u})} \right\} d\mathbf{u}.$$

Further, given that

$$\int \mathcal{N}(\mathbf{f}|\mathbf{L}_{\mathcal{Y}\mathcal{Z}}[\mathbf{L}_{\mathcal{Z}}]^{-1}\mathbf{u}, \mathbf{L} - \mathbf{Q})\mathbf{f}^T\mathbf{f}d\mathbf{f} = \boldsymbol{\alpha}^T\boldsymbol{\alpha} + \text{tr}(\mathbf{L} - \mathbf{Q}),$$

where $\boldsymbol{\alpha} = \mathbf{L}_{\mathcal{Y}\mathcal{Z}}[\mathbf{L}_{\mathcal{Z}}]^{-1}\mathbf{u}$, the bound can be written as

$$\mathcal{F} \geq \int q(\mathbf{u}) \log \frac{\mathcal{N}(\mathbf{u}|\mathbf{0}, \mathbf{L}_{\mathcal{Z}})e^{-\frac{1}{2}\boldsymbol{\alpha}^T\boldsymbol{\alpha}}}{q(\mathbf{u})}d\mathbf{u} - \frac{1}{2}\text{tr}(\mathbf{L} - \mathbf{Q}).$$

Now if we analytically maximize w.r.t. $q(\mathbf{u})$, under the constraint that $q(\mathbf{u})$ is a distribution, we obtain

$$q(\mathbf{u}) = \frac{\mathcal{N}(\mathbf{u}|\mathbf{0}, \mathbf{L}_{\mathcal{Z}})e^{-\frac{1}{2}\boldsymbol{\alpha}^T\boldsymbol{\alpha}}}{\int \mathcal{N}(\mathbf{u}|\mathbf{0}, \mathbf{L}_{\mathcal{Z}})e^{-\frac{1}{2}\boldsymbol{\alpha}^T\boldsymbol{\alpha}}d\mathbf{u}}.$$

Plugging this optimal $q$ back into the bound, we obtain

$$\mathcal{F} \geq \log \int \mathcal{N}(\mathbf{u}|\mathbf{0}, \mathbf{L}_{\mathcal{Z}}) e^{-\frac{1}{2}\boldsymbol{\alpha}^T \boldsymbol{\alpha}} d\mathbf{u} - \frac{1}{2}\text{tr}(\mathbf{L} - \mathbf{Q}).$$

After computing the Gaussian integral w.r.t. $\mathbf{u}$, the r.h.s. reduces to the logarithm of the DPP bound for the finite case, see Proposition 1.

## B    $\Psi$ matrix for Gaussian kernels

We give here more details on the Gaussian kernel with Gaussian base measure used in the experimental Section 5. We use the notation of Section 5.2. The kernel is

$$L(\mathbf{x}_i, \mathbf{x}_j) = e^{-\sum_{d=1}^{D} \frac{\left(x_{i,d} - x_{j,d}\right)^2}{2\sigma_d^2}},$$

with Gaussian base measure having density

$$\mu'(\mathbf{x}) = \kappa \prod_{d=1}^{D} \frac{1}{\sqrt{2\pi\rho_d^2}} e^{-\frac{1}{2\rho_d^2}(x_d - \mu_d)^2}.$$

In this Gaussian-Gaussian case, the $\Psi$ matrix defined in Proposition 2 can be analytically computed: the $ij$-th element is given by

$$[\Psi]_{ij} = \int_{\mathbb{R}^D} L(\mathbf{z}_i, \mathbf{x}) L(\mathbf{x}, \mathbf{z}_j) d\mu(\mathbf{x}) = \kappa \prod_{d=1}^{D} \frac{e^{-\frac{1}{4}\sigma_d^{-2}(z_{i,d} - z_{j,d})^2 - \frac{\sigma_d^{-2}(\mu_d - \bar{z}_d)^2}{2\sigma_d^{-2}\rho_d^2 + 1}}}{\left(2\sigma_d^{-2}\rho_d^2 + 1\right)^{\frac{1}{2}}}, \qquad (20)$$

where $\bar{z}_d = \frac{z_{i,d} + z_{j,d}}{2}$.

## Footnotes

[1] Notice that $\int \mathcal{N}(\mathbf{f}|\mathbf{0}, \mathbf{L})e^{-\frac{1}{2}\mathbf{f}^T\mathbf{f}}d\mathbf{f} = (2\pi)^{n/2}\int \mathcal{N}(\mathbf{f}|\mathbf{0}, \mathbf{L})\mathcal{N}(\mathbf{0}|\mathbf{f}, \mathbf{I})d\mathbf{f} = (2\pi)^{n/2}\mathcal{N}(\mathbf{0}|\mathbf{0}, \mathbf{L} + \mathbf{I})$, and (9) follows.

[2]Note that this necessarily happens under fairly weak assumptions: saying that the upper and lower bounds in (4) match when $m$ goes to infinity is saying that the integral of the posterior variance of a Gaussian process with no evaluation error goes to zero as we add more distinct training points.

[3]i.e., for which all points in a realization are distinct.

[4]There is a notion of kernel $K$ for general DPPs [5], but we define $L$ directly here, for the sake of simplicity. The interpretability issues of using $L$ instead of $K$ are the same as for the finite case, see Sections 2 and 5.

[5]We follow the parametrization of [20] for ease of reference.