[Reviews · NeurIPS 2015]

Submitted by Assigned_Reviewer_1

Determinantal point processes (DPPs) have been receiving a significant amount of recent attention in the machine learning literature, due to their ability to model repulsion within realizations of a set of points.

This repulsion is specified via a positive definite kernel function.

One challenge, however, is learning the appropriate parameters of this kernel, given example point realizations.

This estimation is difficult due to a difficult-to-compute normalization constant that, in effect, sums over all possible cardinalities and point configurations.

A recent proposal [7] has provided a way to do this by observing that the spectrum can be estimated and used to provide incremental upper and lower bounds, resulting in a provably-correct scheme for "retrospective" Markov chain Monte Carlo (MCMC) of the kernel parameters.

The present paper proposes a different type of bound that can also be used for MCMC inference, but that does not require estimation of the eigenvalues.

This approach also lends itself to a variational approach to learning, and the paper draws connections to inducing-point methods to scalable inference in Gaussian process models.

This is a technically strong and well-written paper.

I was not aware of the inequality that provides the lower bound in Proposition 1 and it seems like an excellent fit for this problem.

It is a clever way to avoid the estimation issues of [7].

I love the connections to GP inducing points, as well as the ability to now perform variational inference.

Overall this paper was a pleasure to read.

I have two technical concerns that I would like to see addressed, however.

First, although I like this approach very much, I do not find it all that compelling that it is a huge computational win to replace power iterations with nonlinear optimization problems of increasing dimension.

Second, it is not obvious to me that simple increasing of the cardinality is guaranteed to result in convergence of the bounds, because of the coupled optimization problem.

Although I am not certain, I believe this optimization is highly non-convex and so one needs to increase cardinality and also ensure that a global minimum is achieved.

This is something that should be addressed directly, because it seems that a local minimum could prevent the procedure from being able to bound away the Bernoulli threshold and make it impossible to take an MCMC step.

Presentation Issues:

- P7 Figure 1: Please include axis labels for the figures.

- The proof of the proposition is the main thing that makes this

paper possible and is the central insight.

I would've liked to see

it in the main body of the paper and not the index.

- P7 L361: "... the variational lower is ..."
Summary: A nice paper that uses a clever trick to bound the partition function of a determinantal point process, leading to potentially faster inference and connections to other kinds of models.

Submitted by Assigned_Reviewer_2

The authors propose novel upper and lower bounds on the matrix determinant, with applications to bounding the likelihood of a determinantal point process. The main idea is simple, but the paper is well executed and enlightening, and paves the way for several potential improvements.

The authors propose a creative use of the pseudo-inputs to bound functionals of a positive definite kernels. The proposed problem is timely, and will be of interest to several in the community.

Such an approach is quite general with potential applications beyond determinantal point processes. The experimental evaluation is well constructed to elucidate several key points, including failures of a variational approximation under certain conditions. I found the selected application area interesting and highly relevant to the studied problem.

Minor comments:

- It is unclear why the issue of parameterizing K (instead of L) features so prominently in the abstract and the rest of the paper, as the authors mainly focus on describing the problem, and make no efforts to address it.

- If accepted, I suggest that the authors prepare a standard supplement, and not just upload an extended version of the paper as the supplement.

Suggestions for future work:

- On the other hand, the discussion of the interpretability of the K parameterization makes such an approach even more appealing. Could the authors consider bounding the determinant directly in terms of the K matrix? What is the scalability of such an approach if using the pseudo-input method?
Summary: The authors propose novel upper and lower bounds on the matrix determinant, with applications to bounding the likelihood of a determinantal point process. The main idea is simple, but the paper is well executed and enlightening, and paves the way for several potential improvements.

Submitted by Assigned_Reviewer_3

The paper proposes approximate inference techniques for determinantal point process (DPP) that do not require spectral knowledge of the kernel defining the DPP. Instead, inducing inputs are utilized to bound the likelihood, which reduces the computational cost. Both variational inference and approximate Markov chain Monte Carlo (MCMC) techniques are described and their efficiency is demonstrated empirically.

I believe there is merit to this paper, however I found the method description rather rough and empirical evaluation not thorough enough. A few points that may help clarify:

1) How does the computational complexity and overall compute time of the two proposed methods compare?

2) How do the proposed methods compare to other alternatives, both approximate and exact, in terms of quality of approximation and computation time?

3) Empirical results on the toy data in Section 5.1 show that parameter estimation error using variational inference is too large, and resorts to approximate MCMC instead. In what settings does the proposed variational inference expected to work and when is it expected to fail? Why?

4) A 13-point sample is generated for the toy example. How do the methods perform when the inducing inputs are actually these sampled points?

5) Is there a rule of thumb to follow when selecting a) the number of inducing points, b) placing the initial inducing points?

Minor comments:

The matrices K and L are referred to throughout the introduction without explaining what they are. Briefly mentioning the marginal kernel and L-ensembles may make it easier to follow for an uninformed reader.

There is discussion about the advantages of directly learning the marginal kernel parameters throughout the paper, but this is not the approach taken. I believe some of it could be cut back since it does not contribute to the rest of the paper.

Summary: Inducing inputs are utilized to bound the DPP likelihood, which reduces the computational cost. Both variational inference and approximate Markov chain Monte Carlo (MCMC) techniques are described and their efficiency is demonstrated empirically.

I believe there is merit to this paper, however I found the method description rather rough and empirical evaluation not thorough enough. If accepted, I strongly encourage the authors to improve the write-up.

Author Feedback
Author rebuttal: We thank the reviewers for their comments and suggestions.

n = number of objects in the case of a finite DPP
m = number of inducing inputs

We first answer Reviewer 1's comments. First, for finite DPPs with a dense kernel matrix, the power method requires iterative multiplications of a vector with an n x n dense matrix. Iterating p times leads to an overall cost of O(p n^2). Further, the power method requires O(n^2) storage to save the full Gram matrix. As far as we know, it is not possible to apply the power method to large, say 50k x 50k, dense matrices. In contrast, our variational inducing point approach makes this possible, since cost is now O(n m^2) time and O(n m) storage, assuming m < n. Indeed, the heavy computations are associated with the cross n x m kernel matrix between all points in the set and the m inducing points. The cost of one iteration is less clear for our MCMC approach, since m is now a random variable, but toy experiments suggest what matters is the dimension of the ambient space, not the number n of items. Note also that similarly to any retrospective MCMC scheme, an "unlucky" draw of uniform random variable can always result in an arbitrarily costly MCMC iteration.

Second, we agree that optimizing the locations of a fixed set of inducing inputs may only lead to local optima, but at least augmenting the number m of inducing inputs will end up making the MCMC acceptance decision possible if the kernel is smooth enough. Indeed, saying that the upper and lower bounds in (15) match when m goes to infinity is saying that the integral of the posterior variance of a GP with no evaluation error goes to zero as we add more distinct training points. A fine enough grid will thus always do, for instance, and hopefully optimizing the inputs further helps. This is what we observe in the experiments of Section 5.

We now answer Reviewer 3's questions.
1) and 2) please see first paragraph above for costs. In terms of approximation error, we emphasize that our retrospective MCMC scheme is exact, like that of reference [7]: it makes the same acceptance decisions as vanilla MCMC would make if we could evaluate the likelihood. For our variational approach, we experimentally assess the approximation error on a real example in Section 5.2. Unfortunately, we have no theoretical result on the approximation error of the variational approach yet, save the bounds of Propositions 1 and 2.

3) It is difficult to say yet when the variational approach will fail, but there are sanity checks for the user. For example, we illustrate one particular failure in Section 5.1, which is related to having only a point estimate and a flat likelihood. We observed inconsistent parameter estimation when we repeatedly executed the variational optimization routine. To the user, this is a red flag, which is why we advocated using a more costly but exact MCMC approach in this section, which confirmed the unidentifiability of one particular parameter that previously affected the overall estimation.

4) The variational approach failed in that case, even with more inducing points than the 13 data points, see Section 5.1. The MCMC approach automatically selects the number of inducing points, and it consistently required between 40 and 80 inducing points for 10k MCMC iterations here, which is more than the 13 data points. Finally, the data points are a noisy version of what the inducing points should be: the latter, once optimized, look like an "ideal" sample from the underlying DPP, i.e. a well-spread sample from the base measure, see Figures 1a and 4.

5) In the experiment of Section 5.2, we show in Figure 3 the dependency on m of the parameter estimates and the corresponding likelihood bounds. In practice, a user may try increasing values of m and stop when parameter estimates stop varying: here m = 600 could be a good choice depending on one's tolerance. We also recommend repeating the experiments to detect issues as mentioned above in 3). For the MCMC approach, the choice of m is not left to the user, though this is at the price of a higher computational cost. Indeed, at a given MCMC iteration, m is a random variable, and it depends nontrivially on the uniform random variable drawn in the MCMC loop, the data, the kernel and the base measure. We have no precise theoretical result yet that relates m to these quantities, we can only say that in experiments such as in Section 5.1, we only needed at most m=80 points for a 1D example with smooth kernel. As mentioned above, we conjecture the number m selected by MCMC depends on the ambient dimension more than on the number n of items in the finite case, which is good for scalability. Finally, for both methods we initialized the inducing inputs to iid samples from either the base measure or uniform distributions; we observed no significant change, as the optimization routines delivered optima that were independent of the initialization.